# Description of the Three Complete Mitochondrial Genomes of Click Beetles (Coleoptera, Elateridae) with Phylogenetic Implications

**Nan Song ***[ID]**, Xingyu Lin and Te Zhao**

College of Plant Protection, Henan Agricultural University, Zhengzhou 450002, China
* Correspondence: songnan@henau.edu.cn

**Abstract:** The family Elateridae, known as click beetles, is a mega-diverse lineage of Coleoptera. Wireworms are the larval stage of click beetles, which are generalist herbivores and which are recognized as economically important pests of crops. To more effectively control and monitor wireworms, it is crucial to understand the genetics, taxonomy and phylogenetics of Elateridae. Here, we sequenced and characterized three complete mitochondrial genomes (mitogenomes) from the subfamily Elaterinae using a next-generation sequencing approach. In addition, we provided the annotated mitogenomes of the newly sequenced species, namely *Parasilesis musculus* (Candèze, 1873), *Melanotus cribricollis* Candèze, 1860 and *Glyphonyx* sp., and compared their arrangement with other closely related species. The secondary structures of tRNA genes and rRNA genes were predicted. Combined with the published mitogenomes of elaterid species, we reconstructed the phylogenetic framework for Elateridae under maximum likelihood and Bayesian inference methods using nucleotide and amino acid sequence datasets separately. The results from the Bayesian analysis based on the nucleotide dataset PCGRNA including all 37 mitochondrial genes were congruent with previous studies. Within the monophyletic Elateridae, two main clades were recovered. The first clade included Elaterinae and *Melanotus*. The second clade consisted of the remaining subfamilies. Physodactylinae and Cardiophorinae formed a sister group. Agrypninae was monophyletic. A subclade comprised Negastriinae and Dendrometrinae.

**Keywords:** mitogenome; next-generation sequencing; phylogenetic; elaterid beetles

## 1. Introduction

Elateridae is the largest family within Elateroidea, with more than 11,000 described species in the world [1,2]. The insects of Elateridae are also known as click beetles due to the clicking noise produced when they are seized by a predator. The characteristic clicking mechanism is considered as a defensive strategy [3]. The adults of click beetles have some characteristic morphology, with an elongated and narrow body and the joint formed by the prothorax and meditruncus. The larvae usually live underground and are generalist herbivores. The click beetle larvae are also called wireworms, which feed on the seeds and roots of plants. Some wireworms are economically important pest species, and they can cause severe damage to numerous agricultural crops, including maize, wheat and potato [4,5].

Despite economic importance of Elateridae as significant agricultural pests, the classification of click beetles is unstable, and phylogenetic relationships among main lineages (especially among subfamilies) remain controversial. Originally, Linnaeus established the Elater category, but subsequent research based primarily on morphological data has led to conflicting hypotheses [6–10]. The extreme morphological diversity of elateroids has made it difficult to arrive at a consensus regarding their classification. Some elaterid subfamilies as classically defined have been supported by molecular analyses using few gene fragments [8,11–16]. The widely accepted subfamilies include Elaterinae, Agrypninae,

Dendrometrinae, Cardiophorinae and Negastriinae [13]. However, the positions of several species-poor lineages have been unstable, for example, Lissominae and Oxynopterinae.

The prior molecular studies of Elateridae are based mainly on a single gene [12,16] or several gene fragments in sequences [8,13,14,17]. Sagegami-Oba et al. (2007) recovered Elateridae as paraphyletic based on *18S* rRNA sequences [16]. Bocakova et al. (2007) used four molecular markers (*18S* rRNA, *28S* rRNA, *rrnL* and *cox1*) and recovered a nonmonophyletic Elateridae [17]. In the analyses of Kundrata and Bocak (2011), the embedded position of Drilidae rendered Elateridae nonmonophyletic [8]. The soft-bodied Cebrioninae was recovered as a part of Elaterinae, and the monophyly of Elaterinae was not supported. Kundrata et al. (2014) conducted a comprehensive phylogenetic analysis of the superfamily Elateroidea based on multiple gene sequences, which included ten subfamilies of Elateridae [14]. Their results supported all elaterid subfamilies as monophyletic except for Denticollinae [14]. Kundrata et al. (2016) recovered Elaterinae as a sister group of the rest of Elateridae [13]. Agrypninae was sister to a clade comprising Morostomatinae, Dendrometrinae, Cardiophorinae and Negastriinae. The members of Hemiopinae, Lissominae, Thylacosterninae and Pityobiinae were placed within Elaterinae [13]. In prior molecular analyses, there were few gene fragments with short sequence length containing limited phylogenetic information that provided insufficient resolving power for the phylogeny of Elateridae. More recently, Douglas et al. (2021) used anchored hybrid enrichment (AHE) data to investigate the phylogenetic relationships of Elateridae [1]. The results placed the bioluminescent lampyroids within the click beetles. At the subfamily level, their analyses recovered the elaterid subfamilies Elaterinae, Agrypninae, Cardiophorinae, Negastriinae, Pityobiinae and Tetralobinae as monophyletic groups [1].

Next-generation sequencing (NGS) technologies have caused a revolution in biology. In particular, NGS has drastically reduced the cost of whole genome sequencing and provides a more effective method to obtain multiple gene sequences and organelle genomes than the traditional Sanger sequencing. Except for a single known case (*Monocercomonoides* sp. [18]), mitochondria as an important organelle was found in most eukaryotic organisms. The high copy number of mitochondrial genomes (mitogenomes) present in each cell [19] makes it easier to be determined than single-copy genes. In insects, the complete mitogenome contains 37 genes, which encode 13 protein-coding genes (PCGs), 22 transfer RNA genes (tRNAs) and two ribosomal RNA genes (rRNAs) [20]. The typical insect mitogenome has a length of 15,000–18,000 nucleotides in size [20]. With the development of NGS technologies, increasing numbers of insect mitogenomes have been sequenced. Mitogenomes as a source of sequence data have been extensively used for insect phylogenetic analysis. Prior mitogenome studies have shown the utility of the mitogenome in species identification and phylogenetic analysis of click beetles [21–24]. However, the numbers of mitogenomes available for the group are very limited. Only 34 mitogenome sequences for 31 species of Elateridae have been published (GenBank, January 2023). The lack of the representative click beetle mitogenome hinders our ability to more thoroughly investigate the systematics and diversification within this group.

In this work, we newly sequenced three mitogenomes of click beetles, *Parasilesis musculus* (Candèze, 1873) [25], *Melanotus cribricollis* Candèze, 1860 [26] and *Glyphonyx* sp., using a NGS approach. The new mitogenome sequences were merged with the already existing data to create a series of mitogenome data matrices for the purpose of phylogenetic analyses of Elateridae. We addressed the controversial taxonomical questions at the subfamily level, with an attempt to make sense of the systematics of click beetles.

## 2. Materials and Methods

### 2.1. Specimen Collection and DNA Extraction

Adult specimens of *P. musculus*, *M. cribricollis* and *Glyphonyx* sp. were collected from a wheat field in Zhengzhou, Henan province, China. They were stored in 100% ethanol and maintained at 4 °C until DNA extraction. Voucher specimens were deposited in the Entomological Museum of Henan Agricultural University. Specimens were identified

based on the external morphology of adults by following the keys in monographs or in the literature (e.g., [27,28]), and by comparison with the pictures in databases (e.g., MCZBASE: The Database of the Zoological Collections, and BOLD Systems: Management & Analysis—Identification). In addition, molecular sequence data were produced to confirm the identity of the species used. Genomic DNA was isolated from the thoracic muscles using the TIANamp Genomic DNA Kit (Tiangen Biotech (Beijing, China) Co., Ltd.), according to the manufacturer protocols.

*2.2. Genome Sequencing, Assembly and Annotation*

In this study, we followed the protocol by Gillett et al. (2014) to reconstruct the mitogenomes from NGS data [29]. Similar amounts of genomic DNA for each click beetle were mixed with other distantly related insect species. Prior to genome sequencing, we sequenced three mitochondrial gene sequences of *cox1*, *cob* and *rrnS* by using PCR amplification and Sanger sequencing. These mitochondrial gene sequences were used as baits to search for target mitogenomes from pooled DNA samples.

Genome sequencing was conducted using the Illumina HiSeq X Ten platform by Beijing Novogene Bioinformatics Technology Co., Ltd. (Beijing, China), with the strategy of 150 bp paired-end reads. Paired-end sequencing libraries with an average insert size of 350 bp were prepared using Illumina TruSeqTM DNA Sample Prep Kit (Illumina, San Diego, CA, USA). No less than 20 Gb of raw paired reads was produced for each library. Raw NGS reads were filtered with NGS QC Toolkit [30]. The remaining clean data were used for the subsequent genome assembly. De novo assembly was performed using IDBA-UD v. 1.1.1 [31]. IDBA-UD assemblies were constructed using an initial *k*-mer size of 41, an iteration size of 10, and a maximum *k*-mer size of 91.

The mitogenome sequences obtained from NGS data were annotated with MITOS [32]. Gene boundaries were refined by alignment with closely related species. tRNA genes were inferred with MITOS [32] and ARWEN [33] programs. The secondary structures of tRNA genes were redrawn manually in Adobe Illustrator CS. The secondary structures of rRNA genes were predicted with reference to *Gonocephalum outreyi* [34]. The circular mitogenome maps of three species were generated using Mtviz (http://pacosy.informatik.uni-leipzig.de/mtviz, accessed on 26 February 2023) (Figure 1). The newly sequenced mitogenomes have been submitted to GenBank under the accession numbers of OQ475941−OQ475943.

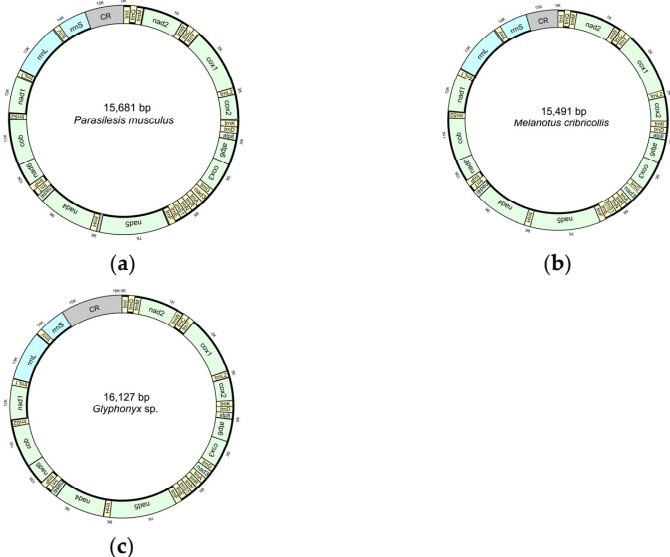

(a)        (b)

(c)

**Figure 1.** Organization of the new mitogenomes. The coding strand is indicated by a thick line. For the color of background, green indicates the protein-coding genes, yellow indicates the tRNA genes, blue indicates the rRNA genes, and gray indicates the control region. Abbreviations for mitochondrial genes follow those in MITOS web.

*2.3. Sequence Alignment*

Thirty-two Elateridae species including three newly sequenced species were included as ingroups. Two species from Lampyridae, three species from Rhagophthalmidae and one species from Omalisidae were selected as outgroups (Table 1). The 13 PCGs were individually aligned by MUSCLE implemented in MEGA 11 [35]. Firstly, the PCGs were translated into amino acid sequences using the invertebrate mitochondrial genetic code and then were aligned based on their amino acid sequence. The alignments were back-translated into the corresponding nucleotide sequences and trimmed manually based on the codons to remove the ambiguous regions. The tRNA and rRNA genes were aligned with MAFFT using E-INS-i iterative method separately [36]. The resulting alignments were concatenated with FASconCAT-G_v1.04 [37] to create the following datasets: (1) PCG_aa, amino acid sequences of 13 PCGs; (2) PCG_nt, nucleotide sequences of 13 PCGs; and (3) PCGRNA, nucleotide sequences of 13 PCGs, 22 tRNAs and 2 rRNAs.

**Table 1.** Taxa included in this study.

| Family | Subfamily | Species | Accession Number | Reference |
|---|---|---|---|---|
| Elateridae | Agrypninae | *Agrypnus* sp. YD-2019 | MN370897 | [38] |
| Elateridae | Agrypninae | *Cryptalaus larvatus* (Candèze, 1874) | NC_047286 | [39] |
| Elateridae | Agrypninae | *Cryptalaus Yamato* (Nakane, 1957) | NC_046689 | Unpublished |
| Elateridae | Agrypninae | *Hapsodrilus ignifer* (Germar, 1841) | NC_030058 | [40] |
| Elateridae | Agrypninae | *Ignelater luminosus* (Illiger, 1807) | MG242621 | [41] |
| Elateridae | Agrypninae | *Pyrearinus termitilluminans* (Candèze, 1863) | NC_030059 | [40] |
| Elateridae | Agrypninae | *Pyrophorus divergens* Eschscholtz, 1829 | NC_009964 | [24] |
| Elateridae | Cardiophorinae | *Dicronychus cinereus* (Herbst, 1784) | KX087283 | Unpublished |
| Elateridae | Cardiophorinae | *Dicronychus* sp. DIC01 | JX412848 | Unpublished |
| Elateridae | Dendrometrinae | *Anostirus castaneus* (Linnaeus, 1758) | KX087237 | Unpublished |
| Elateridae | Dendrometrinae | *Athous haemorrhoidalis* (Fabricius, 1801) | KT876881 | [42] |
| Elateridae | Dendrometrinae | *Campsosternus auratus* Drury, 1773 | MZ727583 | Unpublished |
| Elateridae | Dendrometrinae | *Limonius californicus* (Mannerheim, 1843) | NC_028541 | [43] |
| Elateridae | Dendrometrinae | *Limonius minutus* (Mannerheim, 1843) | KX087306 | Unpublished |
| Elateridae | Dendrometrinae | *Pectocera* sp. | NC_061359 | Unpublished |
| Elateridae | Dendrometrinae | *Pheletes quercus* (Olivier, 1790) | KX087332 | Unpublished |
| Elateridae | Elaterinae | *Adrastus rachifer* (Fourcroy, 1785) | KX087232 | Unpublished |
| Elateridae | Elaterinae | *Agriotes hirayamai* Miwa, 1934 | MG728108 | [44] |
| Elateridae | Elaterinae | *Agriotes lineatus* (Linnaeus, 1758) | OW618681 | Unpublished |
| Elateridae | Elaterinae | *Agriotes obscurus* (Linnaeus, 1758) | KT876879 | [42] |
| Elateridae | Elaterinae | ***Glyphonyx* sp.** | OQ475943 | This study |
| Elateridae | Elaterinae | *Ludioschema sulcicolle* (Candèze, 1878) | NC_053929 | [21] |
| Elateridae | Elaterinae | *Ludioschema vittiger* (Heyden, 1887) | MN306531 | [45] |
| Elateridae | Elaterinae | ***Melanotus cribricollis* (Candèze, 1860)** | OQ475941 | This study |
| Elateridae | Elaterinae | *Melanotus villosus* (Geoffroy, 1785) | KT876904 | [42] |
| Elateridae | Elaterinae | ***Parasilesis musculus* (Candèze, 1873)** | OQ475942 | This study |
| Elateridae | Elaterinae | *Sericus brunneus* (Linnaeus, 1758) | KX087344 | Unpublished |
| Elateridae | N/A | *Elateridae* sp. 2 ACP-2013 | MH789726 | [46] |
| Elateridae | N/A | *Elateridae* sp. GENSP01 | JX412817 | Unpublished |
| Elateridae | Negastriinae | *Negastrius sabulicola* (Boheman, 1852) | KX087320 | Unpublished |
| Elateridae | Physodactylinae | *Teslasena femoralis* (Lucas, 1857) | KJ938491 | Unpublished |
| Elateridae | Tetralobinae | *Sinelater perroti* (Fleutiaux, 1940) | NC_065395 | Unpublished |
| Lampyridae | Luciolinae | *Abscondita cerata* (Olivier, 1911) | MW751423 | [47] |
| Lampyridae | Luciolinae | *Curtos fulvocapitalis* (Jeng and Sato, 1998) | NC_058281 | Unpublished |
| Omalisidae | N/A | *Omalisus fontisbellaquei* (Müller, 1764) | JX412744 | Unpublished |
| Rhagophthalmidae | N/A | *Rhagophthalmus giganteus* (Fairmaire, 1888) | MK292104 | [48] |
| Rhagophthalmidae | N/A | *Rhagophthalmus lufengensis* Li & Ohba (Li et al., 2008) | DQ888607 | [49] |
| Rhagophthalmidae | N/A | *Rhagophthalmus ohbai* (Wittmer, 1994) | NC_010964 | [49] |

Note: Bold indicates the species newly sequenced in this study.

### 2.4. Phylogenetic Reconstruction

Phylogenetic analyses were performed under maximum likelihood (ML) and Bayesian inference (BI) methods. ML tree reconstructions were carried out using IQ-TREE web server [50]. For the alignments including PCGs, the data were partitioned by codon and gene types. For the amino acid sequences, the data were partitioned by gene type. The best-fitting substitution models for each partition were selected using ModelFinder [51]. Branch support (BS) analysis was conducted using ultrafast bootstrap [52], with 10,000 replicates.

Bayesian tree searches were conducted with PhyloBayes-MPI 1.8 [53]. The CAT-GTR model was used for nucleotide sequence analyses, while the CAT-mtArt model was used for amino acid sequence analysis. Two independent runs were performed for each dataset. The program *bpcomp* was used to check the convergence between runs. When the largest discrepancies among partitions (maxdiff) fell below 0.1, good runs were considered to be obtained. The consensus tree was built by discarding the first 20% of trees. The posterior probability (PP) values were calculated for assessing branch support.

We used four-cluster likelihood-mapping approach (FcLM) to study the phylogenetic signal in our datasets. FcLM analyses were conducted with IQ-TREE version 2.2.0 [50]. Taxon clusters were defined to test the conflicting topologies from different datasets.

## 3. Results

### 3.1. Characteristics of the New Mitogenomes

The three new mitogenomes contained the entire set of 37 mitochondrial genes, namely 13 PCGs, 22 tRNAs and two rRNAs (Figure 1 and Table S1). In addition, a non-coding control region (also called the AT-rich region) was identified between *rrnS* and *trnI*. The gene order in the three mitogenomes is identical to the putative ancestral insect mitogenome [20]. The lengths of mitogenomes were 15,491 bp for *M. cribricollis*, 15,681 bp for *P. musculus*, and 16,127 bp for *Glyphonyx* sp. The length variation in the mitogenomes was mainly due to the control region. The lengths of the control region were 846 bp for *M. cribricollis*, 996 bp for *P. musculus*, and 1438 bp for *Glyphonyx* sp. Overall, the three mitogenomes were very compact. The intergenic spacer sequences of the mitogenomes totaled 116 bp for *M. cribricollis*, 143 bp for *P. musculus*, and 130 bp for *Glyphonyx* sp. The gene overlaps were 181 bp (*M. cribricollis*), 237 bp (*P. musculus*) and 107 bp (*Glyphonyx* sp.).

Similar nucleotide compositions were shared by the mitogenomes. They were biased toward A and T, with 72.0–75.1% A + T content. For the protein-coding genes, the range of variation of AT% spanned from 70.0% (*P. musculus*) to 74.0% (*M. cribricollis*), with the average value equal to 72.0%. The tRNA and rRNA genes had higher A + T content. The A + T content of tRNA genes ranged from 77.2% (*P. musculus*) to 78.8% (*M. cribricollis*), and that of rRNA genes ranged from 75.5% (*P. musculus*) to 75.7% (*Glyphonyx* sp.). The control region exhibited the highest AT%, with the range of variation spanning from 77.6% (*P. musculus*) to 82.8% (*Glyphonyx* sp.). GC-skews of the plus strand were $-0.264$ (*M. cribricollis*), $-0.212$ (*P. musculus*) and $-0.23$ (*Glyphonyx* sp.). All three mtDNAs displayed negative GC skews. This result indicated the occurrence of more Gs than Cs in the mitogenomes.

In each of the mitogenomes, most PCGs started with the typical codon ATN (ATT, ATG, and ATA), except for *nad1* used for TTG. Most terminated with the canonical stop codons (TAG and TAA), while incomplete stop codons (T or TA) were inferred for *cox2 and cox3* (*P. musculus*, *M. cribricollis* and *Glyphonyx* sp.), *nad5* (*P. musculus* and *M. cribricollis*), and *nad4* (*P. musculus* and *Glyphonyx* sp.). The relative synonymous codon usage (RSCU) index suggested a strong trend toward A and T in the third codon positions (Figure 2). In the three species, Ile, Phe, Leu, Met and Ser were among the most frequently used amino acids (Table 2).

The length of a single tRNA gene ranged from 61 to 71 bp. The three click beetles that were newly sequenced had largely identical secondary structures of tRNA genes. Except for *trnS1*, all the other tRNAs could be folded into a typical cloverleaf structure (Figures 3, S1 and S2). In the three mitogenomes, *trnS1* had an incomplete DHU arm.

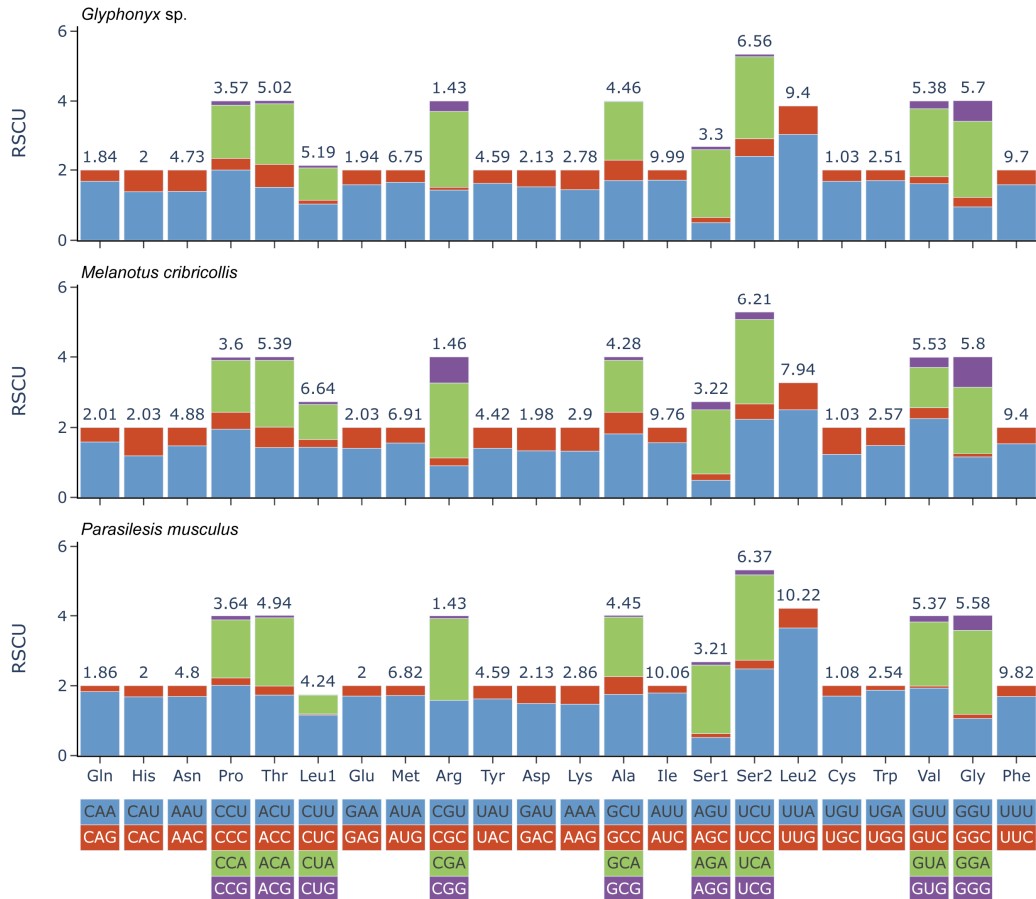

**Figure 2.** Codon usage of the 13 mitochondrial protein-coding genes of *Glyphonyx* sp., *Melanotus cribricollis* and *Parasilesis musculus*. RSCU: relative synonymous codon usage. Codon families are indicated on the X axis and frequency of RSCU on the Y axis.

**Table 2.** Amino acid usage in the new mitogenomes.

| AA | *Parasilesis musculus* | | *Melanotus cribricollis* | | *Glyphonyx* sp. | |
|---|---|---|---|---|---|---|
| | **Count** | **%** | **Count** | **%** | **Count** | **%** |
| Ala(A) | 165 | 4.45 | 158 | 4.28 | 165 | 4.46 |
| Cys(C) | 40 | 1.08 | 38 | 1.03 | 38 | 1.03 |
| Asp(D) | 79 | 2.13 | 73 | 1.98 | 79 | 2.13 |
| Glu(E) | 74 | 2 | 75 | 2.03 | 72 | 1.94 |
| Phe(F) | 364 | 9.82 | 347 | 9.4 | 359 | 9.7 |
| Gly(G) | 207 | 5.58 | 214 | 5.8 | 211 | 5.7 |
| His(H) | 74 | 2 | 75 | 2.03 | 74 | 2 |
| Ile(I) | 373 | 10.06 | 360 | 9.76 | 370 | 9.99 |
| Lys(K) | 106 | 2.86 | 107 | 2.9 | 103 | 2.78 |
| Leu2(L2) | 379 | 10.22 | 293 | 7.94 | 348 | 9.4 |
| Leu1(L1) | 157 | 4.24 | 245 | 6.64 | 192 | 5.19 |
| Met(M) | 253 | 6.82 | 255 | 6.91 | 250 | 6.75 |
| Asn(N) | 178 | 4.8 | 180 | 4.88 | 175 | 4.73 |
| Pro(P) | 135 | 3.64 | 133 | 3.6 | 132 | 3.57 |
| Gln(Q) | 69 | 1.86 | 74 | 2.01 | 68 | 1.84 |
| Arg(R) | 53 | 1.43 | 54 | 1.46 | 53 | 1.43 |
| Ser2(S2) | 236 | 6.37 | 229 | 6.21 | 243 | 6.56 |
| Ser1(S1) | 119 | 3.21 | 119 | 3.22 | 122 | 3.3 |
| Thr(T) | 183 | 4.94 | 199 | 5.39 | 186 | 5.02 |
| Val(V) | 199 | 5.37 | 204 | 5.53 | 199 | 5.38 |
| Trp(W) | 94 | 2.54 | 95 | 2.57 | 93 | 2.51 |
| Tyr(Y) | 170 | 4.59 | 163 | 4.42 | 170 | 4.59 |
| codon end in A or T | 3263 | 88.02 | 2906 | 78.75 | 3064 | 82.77 |
| codon end in G or T | 1908 | 51.47 | 1892 | 51.27 | 1841 | 49.73 |

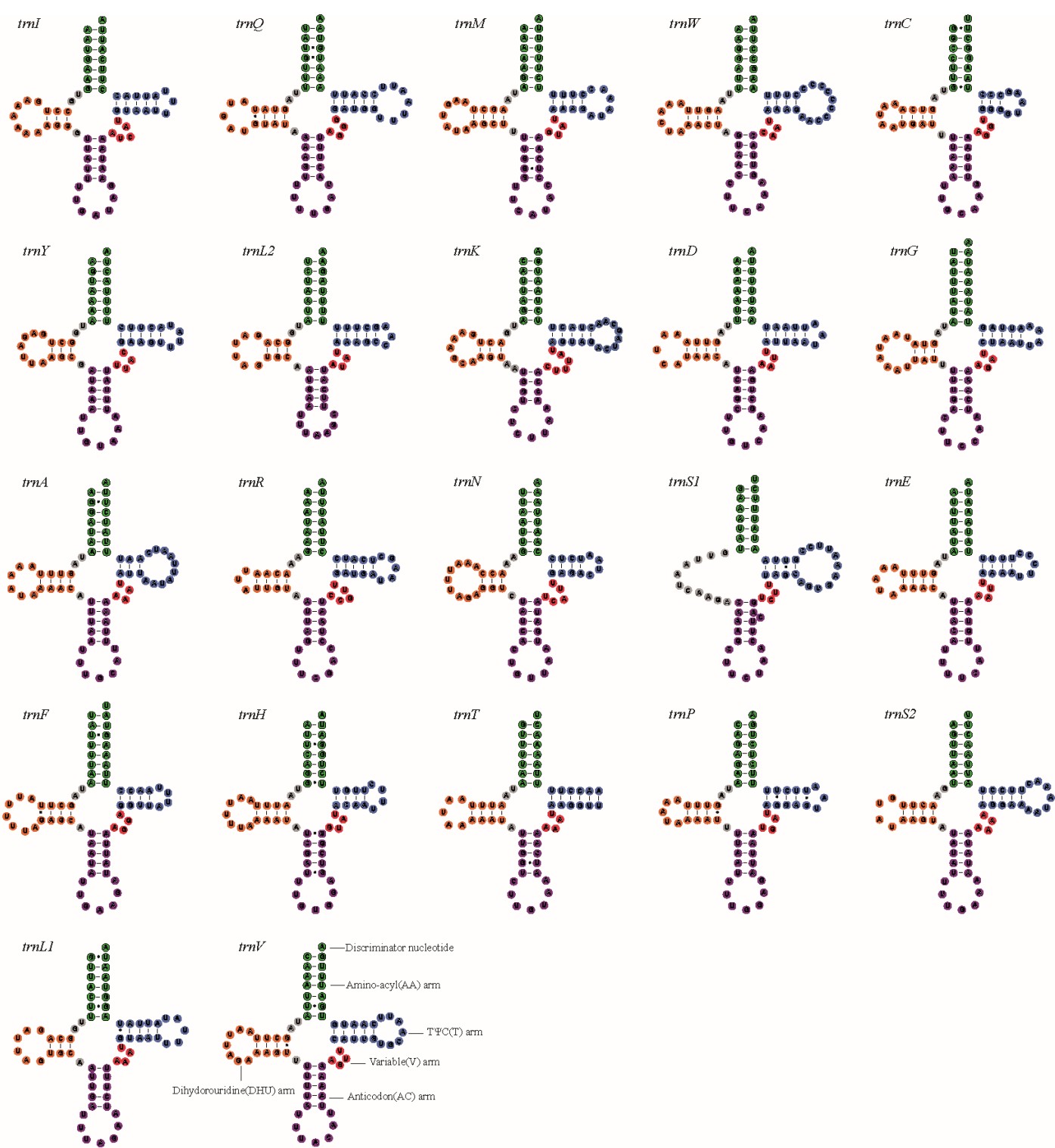

**Figure 3.** The secondary structures of tRNA genes inferred for the mitogenome of *Parasilesis musculus*. Watson–Crick pairs are indicated by lines, and wobble GU pairs are indicated by dots. The non-canonical pairs are not marked.

The lengths of *rrnL* were 1267 bp (*M. cribricollis*), 1301 bp (*P. musculusall*) and 1329 bp (*Glyphonyx* sp.). The three species had the same secondary structures for the *rrnL* and *rrnS* molecules. The secondary structure models for *rrnL* are shown in Figures 4, S3 and S4. *rrnL* had five canonical domains (I–II, IV–VI) and 42 helices. Domain III was missing in three click beetles. The overall structure of *rrnL* was largely in agreement with those

proposed for other beetles (e.g., *G. outreyi*) [34]. There were major differences in the *rrnL* structure between the three click beetles, and *G. outreyi* showed in the domains I, V and VI. Domain I in click beetles included more helices than *G. outreyi*, whereas domains V and VI contained less helices than *G. outreyi*. The lengths of *rrnS* in the three species were 757 bp (*M. cribricollis*), 763 bp (*P. musculusall*) and 762 bp (*Glyphonyx* sp.). Sequence alignments showed that the *rrnS* genes of the three click beetles had basically identical nucleotide sequences. The secondary structure models are shown in Figures 5, S5 and S6. The structure of *rrnS* largely overlapped with previously published structures for *G. outreyi*. The *rrnS* gene was composed of three domains (labeled I, II, III) and 29 helices. Domain II was distinct from that in *G. outreyi*. Domain II in *G. outreyi* contained a long helix (number H673) [34]. However, the newly sequenced click beetles did not have this helix.

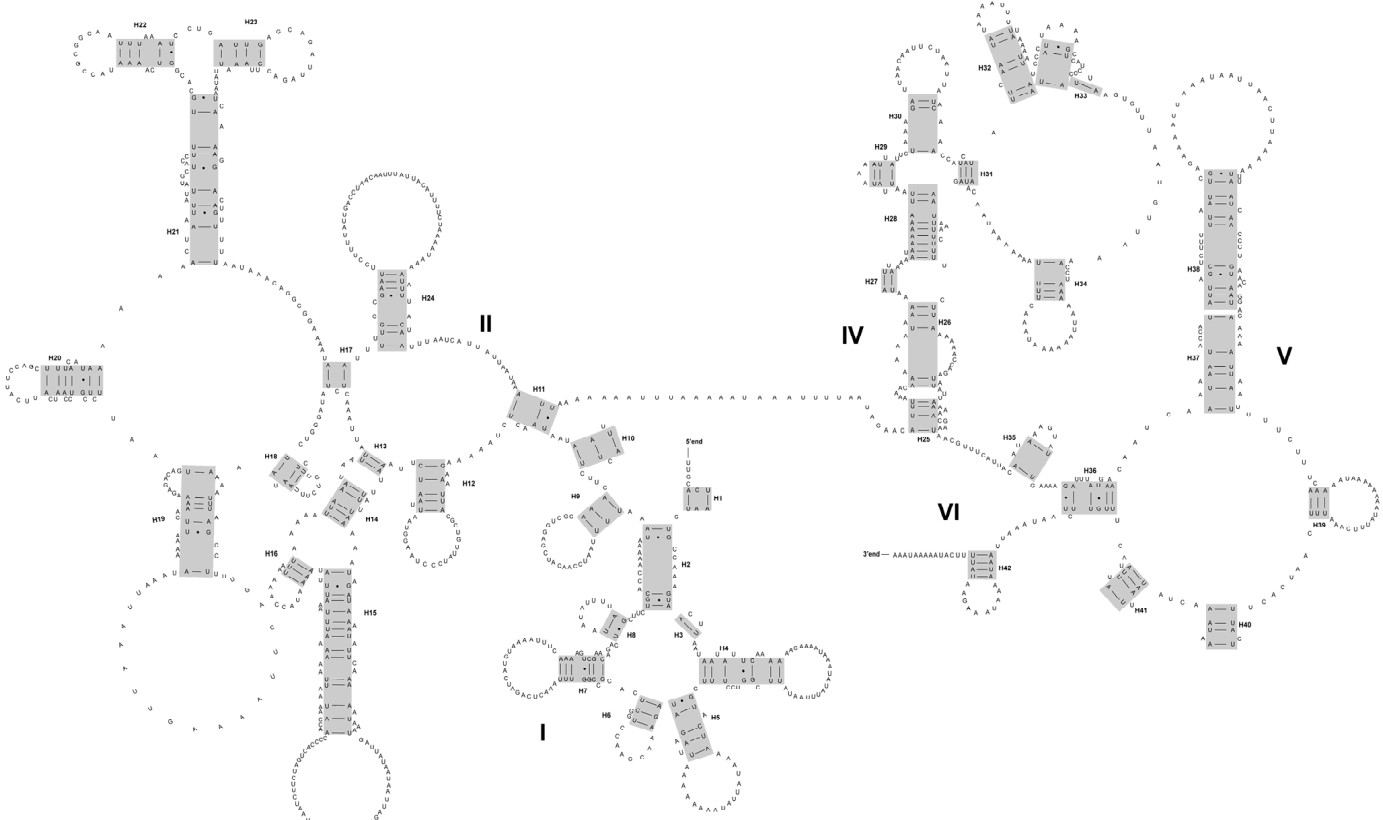

**Figure 4.** Secondary structure of *rrnL* inferred for the mitogenome of *Parasilesis musculus*. Grey boxes indicate the helices. Watson–Crick pairs are indicated by lines, wobble GU pairs are indicated by dots, and the other non-canonical pairs are not marked. The numbers I, II, IV, V and VI represent the five domains in the *rrnL* gene.

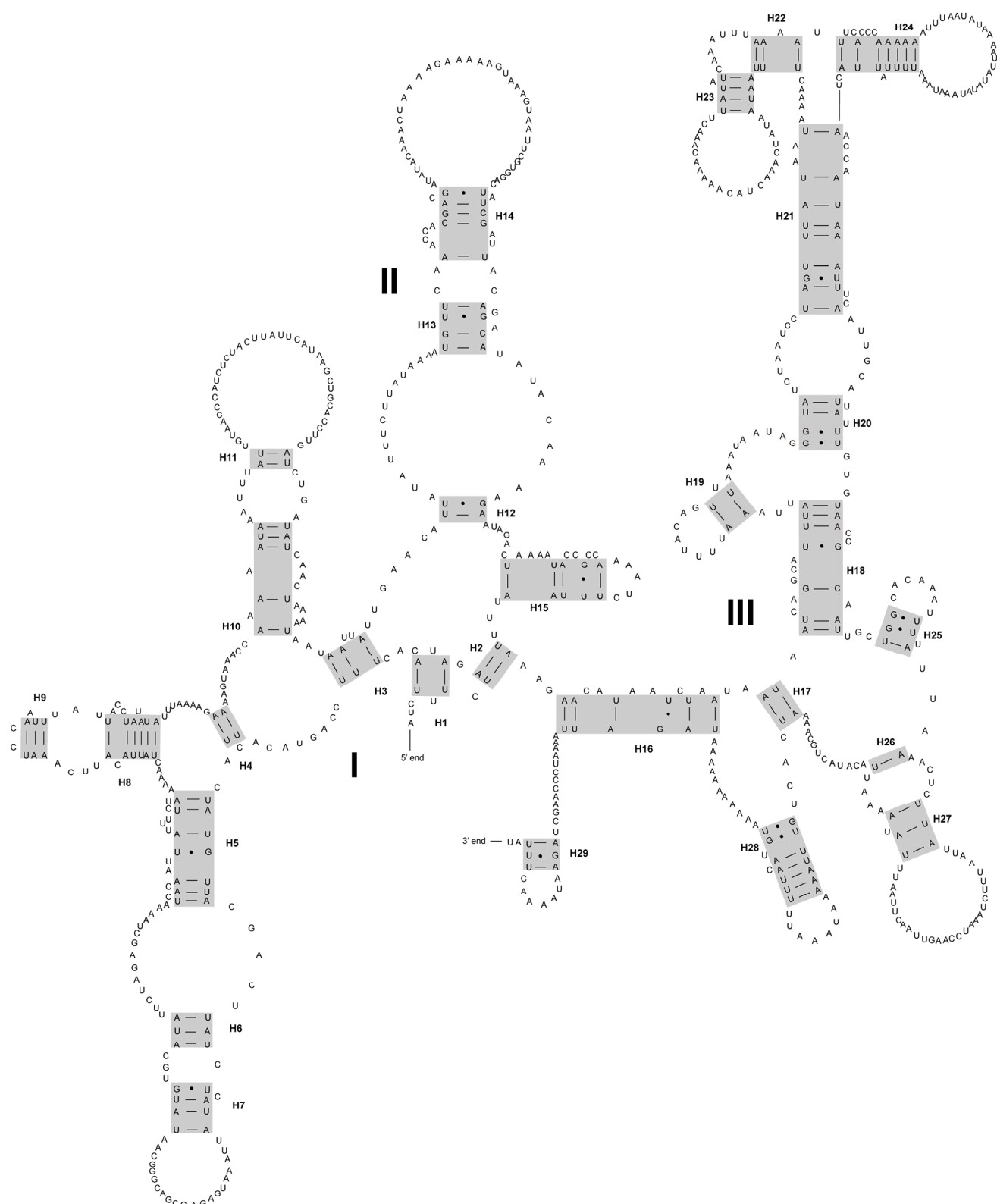

**Figure 5.** Secondary structure of *rrnS* inferred for the mitogenome of *Parasilesis musculus*. Grey boxes indicated the helices. Watson–Crick pairs are indicated by lines, wobble GU pairs are indicated by dots, and the other non-canonical pairs are not marked. The numbers I–III represent the three domains in the *rrnS* gene.

### 3.2. Phylogenetic Inference

The ML reconstructions based on various data matrices produced a largely identical tree topology (Figures 6, S7 and S8). Elateridae were monophyletic in all ML analyses (BS = 88 in PCG_aa-ML tree, BS = 85 in PCG_nt-ML tree, and BS = 90 in PCGRNA-ML tree). The first clade in our phylogeny, sister to the rest of the click beetles, was Tetralobinae. Next, we recovered a clade comprising a sister group of (Physodactylinae + Cardiophorinae). Agrypninae was consistently supported as a monophyletic group. The Elaterinae clade was highly supported (BS = 100 in all ML trees) and inclusive of *Melanotus*.

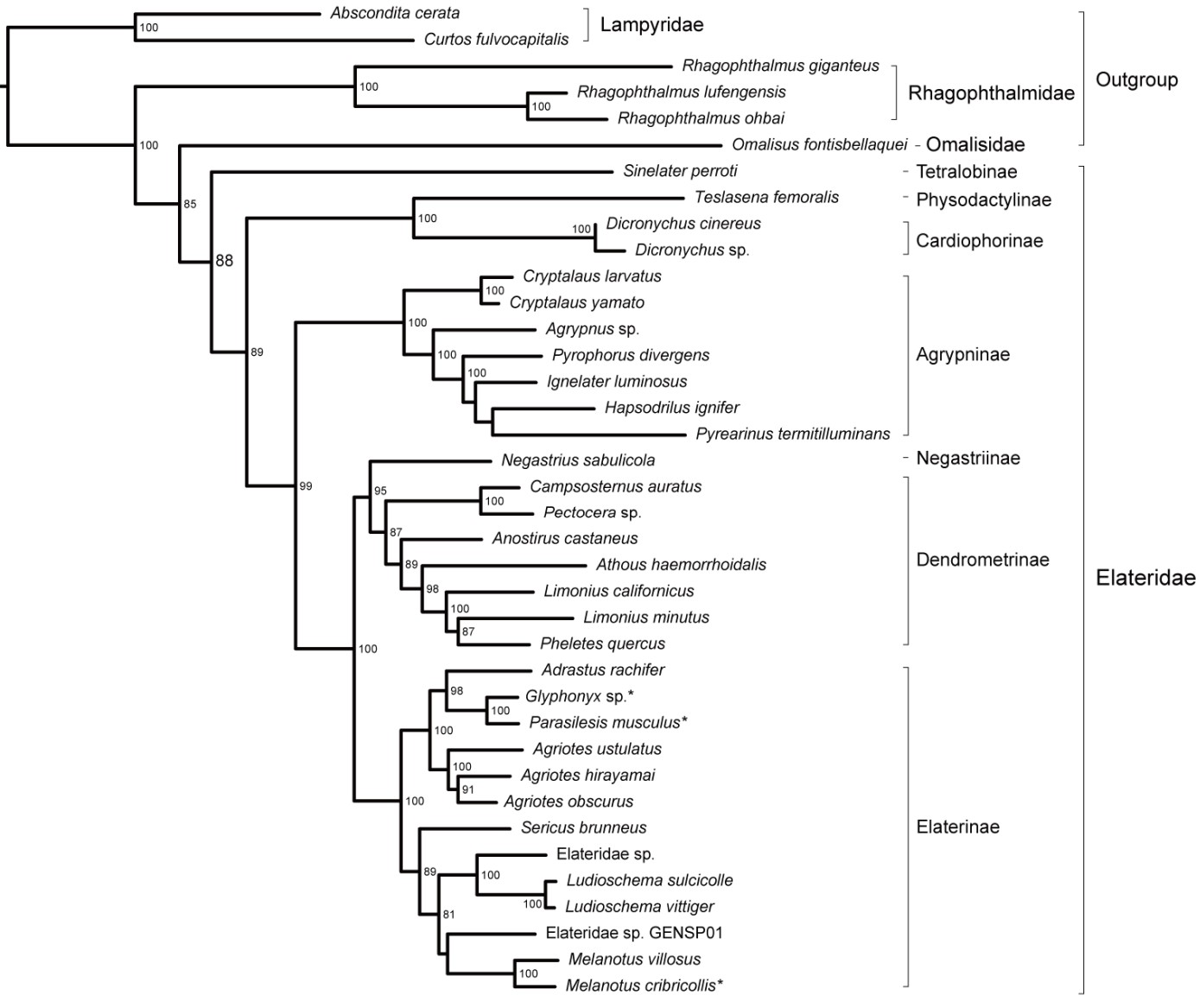

**Figure 6.** ML phylogenetic tree inferred in IQ-TREE using the amino acid sequences of 13 PCGs. Numbers at the nodes are ultrafast bootstrap values (BS > 70). Asterisks indicate the newly sequenced species. Bold indicates the ingroup Elateridae. Scale bar represents substitutions/site.

Dendrometrinae was monophyletic and sister to Negastriinae in the ML tree from the dataset PCG_aa. However, Dendrometrinae was nonmonophyletic in the ML trees from the datasets PCG_nt and PCGRNA. In Figure 7, the FcLM analysis of the amino acid dataset PCG_aa showed 53.8% support for a monophyletic Dendrometrinae, while there was only 26.2% or 20% support for a non-monophyletic Dendrometrinae. In the

dataset PCG_nt, some weaker signal for the monophyly of Dendrometrinae was identified (29.2% of quartets). Despite this result, the support values for alternative topologies were similar. Each grouping had ≥30% support from the dataset PCG_nt. The FcLM result when analyzing the nucleotide dataset PCGRNA showed a stronger signal for the monophyly of Dendrometrinae (36.9% of quartets). Considering the information content of the datasets PCG_aa and PCGRNA, the monophyletic Dendrometrinae was preferred.

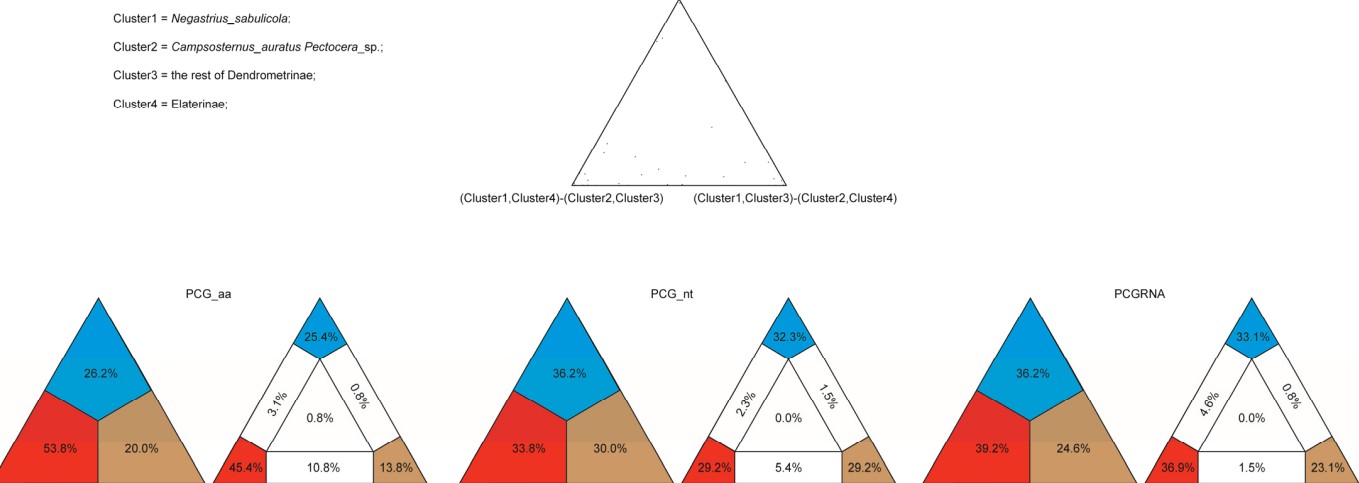

**Figure 7.** Results of the FcLM analyses on different datasets for the phylogenetic hypothesis of Dendrometrinae.

In the BI analyses, only the dataset PCGRNA retrieved Elateridae as monophyletic (Figure 8). In the BI tree from the dataset PCGRNA, two main clades were recovered within Elateridae. The first of these main clades included the subfamily Elaterinae, with *Melanotus* as a subclade in this group. The second clade consisted of the remaining subfamilies. The first subclade included Negastriinae and Dendrometrinae. The second subclade consisted of Agrypninae, which was sister to a clade of (Tetralobinae + (Physodactylinae + Cardiophorinae)). In the BI analyses of PCG_nt (Figure S9) and PCG_aa (Figure S10), Omalisidae had a close relationship with Elateridae or was nested within Elateridae. Despite different branching order inferred from different datasets under the Bayesian inference method, several similar patterns were recovered as ML analyses, such as the sister group relationship between Physodactylinae and Cardiophorinae and the monophyletic Agrypninae and Elaterinae.

Previous studies have suggested that long branches can artificially group together in a phylogenetic reconstruction [54,55]. Long branch attraction artifacts can explain the incorrect branching patterns in the tree. In the current analysis, branch lengths of taxa in the outgroups Rhagophthalmidae and Omalisidae were obviously long in the ML tree from the dataset PCG_aa. Accordingly, we removed the long-branched taxa (*Rhagophthalmus giganteus*, *Rhagophthalmus ohbai* and *Omalisus fontisbellaquei*). As a result, the ML analysis using the reduced taxon amino acid dataset produced the same topology (Figure S11). This suggested that the relationships from ML analysis were not the result of long branch attraction.

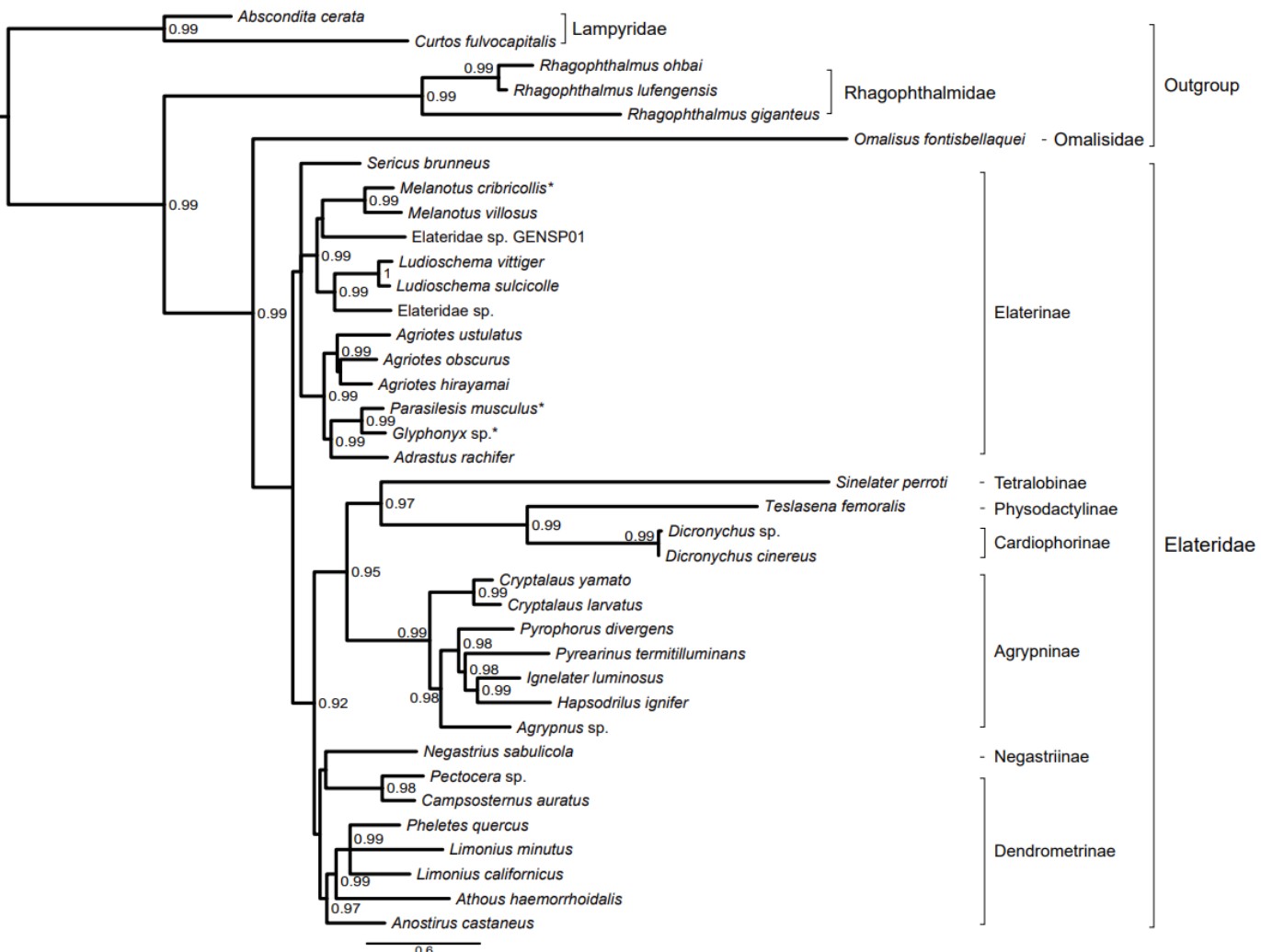

**Figure 8.** Bayesian phylogenetic tree inferred in PhyloBayes-MPI using the nucleotide dataset PCGRNA. Numbers at the nodes in the tree are Bayesian posterior probabilities (PP > 0.9). Asterisks indicate the newly sequenced species. Bold indicates the ingroup Elateridae. Scale bar represents substitutions/site.

## 4. Discussion

This study expanded previous knowledge of mitogenomes in Elateridae by sequencing three representatives of the subfamily Elaterinae. As mentioned, the new mitogenomes were similar in genome organization and nucleotide composition to the presumed ancestral insect [20]. Combined with the existing click beetles' mitogenomes, we reconstructed the phylogeny of Elateridae. Phylogenetic placements of the newly sequenced species were stable across analyses. Mitochondrial phylogenomics robustly resolved *M. cribricollis* as the sister group of another exemplar of the same genus (*M. villosus*). *P. musculus* and *Glyphonyx* sp. were sister groups and were placed within the subfamily Elaterinae. These results show the utility of mitogenomes in the taxonomy and phylogenetics of click beetles.

### 4.1. Monophyly of the Elateridae

The monophyly of Elateridae has been contentious. The family has long been considered as a nonmonophyletic group [6,7,56]. Calder et al. (1993) assigned *Lissomus* Dalman and *Drapetes* Dejean as members of Elateridae [56]. This arrangement was supported by Muona (1995) [7]. Lawrence et al. (2007) [57] indicated that Lissominae (once considered as a separate family Lissomidae, e.g., [58–60]) was a lineage in Elateridae based on combined analysis of morphology and mitochondrial DNA sequence data. In the molecular analysis

of Bocakova et al. (2007), the nested positions of Drilidae and Omalisidae rendered Elateridae paraphyletic [17]. Kundrata and Bocak (2011) placed Drilidae in Elateridae and proposed to classify most Drilidae as a tribe (i.e., Drilini) in the subfamily Agrypninae [8]. Omalisidae was consistently recovered as a sister group of Elateridae [8]. Kusy et al. (2021) also recovered *Drilus* (Drilidae) as a member in Agrypninae using genome-scale data and mitogenome sequence data [61,62]. Douglas et al. (2021) recovered a sister group relationship between Elaterinae and a clade including Oestodinae and lampyroids, which rendered Elateridae as paraphyletic [1]. Currently, no mitogenomes of Drilidae are available; we cannot assess the position of the family. In this study, four of six analyses recovered Omalisidae as the sister group of Elateridae. In the remaining two analyses, Omalisidae was nested within Elateridae. Sequencing additional mitogenomes and employing other types of molecular markers (e.g., whole-genome scale data) are expected to address the question of monophyly of Elateridae.

### 4.2. Subfamily Relationships in Elateridae

The subfamily Agrypninae was consistently supported as monophyletic. This result was in agreement with several previous studies [1,13,16,22,62]. In the BI tree from the dataset PCGRNA, Agrypninae was sister to a clade comprising Tetralobinae, Physodactylinae and Cardiophorinae. They were also sister to a clade including the substantial Dendrometrinae. These subfamily relationships were congruent with the analysis of Kundrata et al. (2016) [13].

The position of Cardiophorinae has been debated in previous studies. Cardiophorinae was once placed as the sister group of the remaining Elateridae [9,63]. Stibick (1979) found Negastriinae as the sister group of Cardiophorinae [6]. Calder et al. (1993) identified a close relationship of Cardiophorinae to Elaterinae [56]. Oba (2007) indicated a close affinity of Negastriinae to Cardiophorinae [64]. Douglas (2011) suggested that members assigned to Physodactylinae or Denticollinae were the closest relatives of Cardiophorinae [10]. Kundrata and Bocak (2011) supported a sister group relationship between Negastriinae and Cardiophorinae [8]. This arrangement was confirmed in the phylogenomic analysis based on AHE data [1]. In our analyses, only a single species representing Negastriinae (i.e., *Negastrius sabulicola*) was included. *N. sabulicola* was nested within Dendrometrinae and isdistantly related to Cardiophorinae. A sister group relationship between Cardiophorinae and Physodactylinae (*Teslasena*) was supported by the present mitogenome data. Douglas (2011) conducted the phylogenetic analyses of Elateridae based on 175 adult morphological characters [10]. Especially, he discussed the placement of Cardiophorinae. In some analyses, a close relationship between Cardiophorinae and *Teslasena* was recovered [10]. A prior mitogenome analysis also supported a sister group relationship between Cardiophorinae and *Teslasena* [62]. Our results were consistent with the two studies [10,62].

The monophyly of the subfamily Dendrometrinae has been challenged [11,13]. Kundrata and Bocak (2011) recovered a nonmonophyletic Dendrometrinae based on the combined analysis of nuclear and mitochondrial gene sequences [8]. They proposed to include the subfamilies Oxynopterinae and Semiotinae in Dendrometrinae [8]. Bocak et al. (2018) included Plastoceridae in Dendrometrinae [15]. In our analyses, amino acid data supported Dendrometrinae as monophyletic, while nucleotide datasets recovered Dendrometrinae to be nonmonophyletic with respect to Negastriinae. FcLM tests showed a more genuine phylogenetic signal for Dendrometrinae in the amino acid dataset PCG_aa and the nucleotide dataset PCGRNA. Thus, our data basically supported the monophyly of Dendrometrinae.

*Melanotus* was once classified as the independent subfamily Melanotinae. However, the monophyly of the clade of Elaterinae + Melanotinae was supported by the analyses from molecular data [16] or morphological data [63,65]. Recently, more systematists have treated *Melanotus* as a tribe in Elaterinae [1,8,12,22,62]. In this study, the Elaterinae including *Melanotus* was supported across analyses.

## 5. Conclusions

In the present study, we sequenced three mitogenomes of click beetles, *P. musculus*, *M. cribricollis* and *Glyphonyx* sp. of which the mitogenomes of *P. musculus* and *Glyphonyx* sp. represented the first from their genera. Details on their structure and sequence characteristics have been presented. These results included size description, genes and nucleotide composition. The mitogenomes ranged from 15,491 to 16,127 bp. These values fall within a normal range for insect mitogenomes. All three mitogenomes contained the typical set of 37 genes and a putative control region. The gene arrangement was identical to the putative ancestral insect mitogenome. Nucleotide composition appeared to be conserved in the mitogenomes of the click beetles examined. The secondary structure models for *rrnL* and *rrnS* of the three species were inferred. This is the first prediction for click beetles. The overall structures of *rrnL* and *rrnS* genes, as demonstrated by the models produced for *P. musculus*, appeared to be similar to those determined for other coleopteran insects. Phylogenetic analyses suggested that a complete mitogenome sequence provided an excellent tool for the phylogenetic relationship inference of click beetles. Different inference methods under various data-coding strategies produced similar tree topological structures. Most of the analyses supported Elateridae as monophyletic. Two main clades were recovered in Elateridae. The first clade included Elaterinae and *Melanotus*. The second clade consisted of the remaining subfamilies. Physodactylinae represented by *Teslasena* was sister to Cardiophorinae. Agrypninae was monophyletic. A subclade comprised Negastriinae and Dendrometrinae. The overall phylogenetic structure of Elateridae was comparable with previous studies.

**Supplementary Materials:** The following supporting information can be downloaded at: https://www.mdpi.com/article/10.3390/taxonomy3020015/s1, Figure S1: The secondary structures of tRNA genes inferred for the mitogenome of *Melanotus cribricollis*; Figure S2: The secondary structures of tRNA genes inferred for the mitogenome of *Glyphonyx* sp.; Figure S3: The secondary structures of *rrnL* genes inferred for the mitogenome of *Melanotus cribricollis*; Figure S4: The secondary structures of *rrnL* genes inferred for the mitogenome of *Glyphonyx* sp.; Figure S5: The secondary structures of *rrnS* genes inferred for the mitogenome of *Melanotus cribricollis*; Figure S6: The secondary structures of *rrnS* genes inferred for the mitogenome of *Glyphonyx* sp.; Figure S7: ML phylogenetic tree inferred in IQ-TREE using the nucleotide sequences of 13 PCGs; Figure S8: ML phylogenetic tree inferred in IQ-TREE using the nucleotide sequences of all 37 mitochondrial genes; Figure S9: Bayesian phylogenetic tree inferred in PhyloBayes-MPI using the nucleotide sequences of 13 PCGs; Figure S10: Bayesian phylogenetic tree inferred in PhyloBayes-MPI using the amino acid sequences of 13 PCGs; Figure S11: ML phylogenetic tree inferred in IQ-TREE using the reduced taxon amino acid dataset; Table S1: Annotation of the new mitogenomes.

**Author Contributions:** N.S. and T.Z. designed the research. N.S., X.L. and T.Z. performed the research and analyzed the data. N.S. and X.L. wrote the paper. All authors have read and agreed to the published version of the manuscript.

**Funding:** This study was funded by the National Natural Science Foundation of China (U1904104). The funders had no role in the study design, data collection and analysis, decision to publish or preparation of the manuscript.

**Data Availability Statement:** The mitogenome sequences newly generated in this study were deposited in GenBank, with the accession numbers of OQ475941–OQ475943.

**Conflicts of Interest:** The authors declare no competing interests in the study.

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
