# Peer review of "Description of the Three Complete Mitochondrial Genomes of Click Beetles (Coleoptera, Elateridae) with Phylogenetic Implications"

_2673-6500, doi:10.3390/taxonomy3020015_

Round 1
Reviewer 1 Report
The beetles of the family Elateridae is very import for the economy and the science. The paper entitled "Three Complete Mitochondrial Genomes of Click Beetles (Coleoptera, Elateridae) and Implications for the Phylogeny of Elateridae" by Song etal newly sequenced three mitogenomes of click beetles and make a comprehensive analysis of the phylogenetic problems of the family Elateridae. It is very informative for understand the evolutionary clew of this interesting group. The paper is well written, and analysis, discussion are also complete. The content also well match the scope of the journal. I suggest the author make several minor revisions for the manuscript before accpeted for publication. My specific comments are as follow:
1. Please give the complete scientific name of the species when first present in the text. please add the author and year. Also add the reference of the original papers.
2. For the species identification, which monograph or publication used for identification of the species? please add the references.
3. For the molecualr experiment, how many data were generated? 10Gb?
4. The circle of mitogenome were only provide for Parasilesis musculus, please add the circles of other two species.
5. The legends for all Figures need to be improved. Most of information are missing. Difficulat to understand. All figures should be self-explanatory.
6. Are there any differences found in the secondary structure of rrnL, rrnS and tRNA among three species? Please make a comparison or discussion.
7. Are there any morphological characters support the relationship between Cardiophorinae and Physodactylinae? Please explain.
8. Lines 275-277. The amino acide data and nucleotide datasets analysis obtained different resutls. Please explain the reason. What is your opinion for this point?
9. I suggest the author prepare section of 5. Conclusion and summarize the main conclusions of this paper.
10. I suggest the author arrange Table S1 in the main body of the manuscript. Please give the complete name of species in the Table, I mean give the author and the year for each species, and add the references in the list. Please add another column to show the references for each sequencces cited, also add the references in the list. This should be strictly complied because it is the normative rule of science.
See the revised pdf for detail.

Author Response
Responses to comments
Dear Reviewer,
Thank you for your very careful review of our paper, and for the comments, corrections and suggestions that ensued. A revision of the paper has been carried out to take all of them into account. And in the process, we believe that the paper has been significantly improved.
In the present “Responses to comments”, we sequentially address all of the points raised in each of the comments made by editor and different reviewers. Please refer to the page number and line number in the main-text file under Word format.
Review1
The beetles of the family Elateridae is very import for the economy and the science. The paper entitled "Three Complete Mitochondrial Genomes of Click Beetles (Coleoptera, Elateridae) and Implications for the Phylogeny of Elateridae" by Song etal newly sequenced three mitogenomes of click beetles and make a comprehensive analysis of the phylogenetic problems of the family Elateridae. It is very informative for understand the evolutionary clew of this interesting group. The paper is well written, and analysis, discussion are also complete. The content also well match the scope of the journal. I suggest the author make several minor revisions for the manuscript before accpeted for publication. My specific comments are as follow:
Response: thanks very much for your comments. Meanwhile, thanks for your good suggestions for improving the analyses and writing of this manuscript.
- Please give the complete scientific name of the species when first present in the text. please add the author and year. Also add the reference of the original papers.
Response: the author and year of species names have been added in this version (P1 L14-15, P2 L89, Table 1).
- For the species identification, which monograph or publication used for identification of the species? please add the references.
Response: The references 27, 28 has been added in this section. Writing of the sentence was also revised (P3 L99-104).
- For the molecualr experiment, how many data were generated? 10Gb?
Response: In our experiment, no less than 20 Gb raw paired reads were produced for each library. P3 L118.
- The circle of mitogenome were only provide for Parasilesis musculus, please add the circles of other two species.
Response: the circles of other two species have been added in this version Figure1.
- The legends for all Figures need to be improved. Most of information are missing. Difficulat to understand. All figures should be self-explanatory.
Response: The legends for all Figures have been improved in this version.
- Are there any differences found in the secondary structure of rrnL, rrnS and tRNA among three species? Please make a comparison or discussion.
Response: The secondary structures of rrnL, rrnS and tRNA of the three newly sequenced species are largely identical. Specially, the sequences of tRNA and rrnS of three species are basically identical, which result in the identical structures. The statements have been added in P6 L224-225, L229-230.
- Are there any morphological characters support the relationship between Cardiophorinae and Physodactylinae? Please explain.
Response: The related sentences have beend added in P6 L348-351.
- Lines 275-277. The amino acide data and nucleotide datasets analysis obtained different resutls. Please explain the reason. What is your opinion for this point?
Response: For this problem, we conducted FcLM analyses to investigate the reason of the conflicting results. The contents have been added in L168-170, Figure 7 in this version, L262-272.
- I suggest the author prepare section of 5. Conclusion and summarize the main conclusions of this paper.
Response: The section of 5. Conclusion has been added in this version. L369-390.
- I suggest the author arrange Table S1 in the main body of the manuscript. Please give the complete name of species in the Table, I mean give the author and the year for each species, and add the references in the list. Please add another column to show the references for each sequences cited, also add the references in the list. This should be strictly complied because it is the normative rule of science. See the revised pdf for detail.
Response: According to your suggestion, Table S1 has been changed to be Table 1 and moved to the main body of the manuscript. The author and the year for each species have been added. Moreover, the associated references have been listed in this table.

Reviewer 2 Report
The title is too long, it should be shortened
The abstract correctly included the manuscript findings
Some changes must be done:
row 51: fragments is not correct, please modify in sequences
row 75: most is too much. There is a single protista without mitochondria. This reference is here essentially useless, since the research is not focused on protista, but at least (if you truly must quote it) modify into: Except for a single known case (citation), mitochondria...
rows 79-80: why the Glyphonys sp. was not identified using barcode? The issue must be discussed. Furthermore, the first time a species is quoted also the author name must be given.
row 98. Which field do you mean?
Figure 1 [and the subsequent ones]: all the legends must be improved, at present they are a bit obscure.
row 185 what is P. musculusall (also row 190)? Why this species and Glyphonys are underlined? Please, carefully check the text for any further misspelling.
Figure 6 and 7, the quality is very low, images must be improved.
Discussion is supported by the findings
Author Response
Responses to comments
Dear Reviewer,
Thank you for your very careful review of our paper, and for the comments, corrections and suggestions that ensued. A revision of the paper has been carried out to take all of them into account. And in the process, we believe that the paper has been significantly improved.
In the present “Responses to comments”, we sequentially address all of the points raised in each of the comments made by editor and different reviewers. Please refer to the page number and line number in the main-text file under Word format.
Review2
The title is too long, it should be shortened
Response: According to your suggestion, the title has been be shortened in this version.
The abstract correctly included the manuscript findings
Some changes must be done:
row 51: fragments is not correct, please modify in sequences
Response: Thanks for your suggestion, the sentence has been corrected.
row 75: most is too much. There is a single protista without mitochondria. This reference is here essentially useless, since the research is not focused on protista, but at least (if you truly must quote it) modify into: Except for a single known case (citation), mitochondria...
Response: This sentence has been revised according to your suggestion.
rows 79-80: why the Glyphonys sp. was not identified using barcode? The issue must be discussed. Furthermore, the first time a species is quoted also the author name must be given.
Response: We conducted the molecular identification for Glyphonys sp. using barcode (cox1). However, the existing sequences in BOLD and NCBI do not match well with our sequence. As a result, this species was only identified to the genus level. The author names of the species have been added in this version (P1 L14-15, P2 L89, Table 1).
row 98. Which field do you mean?
Response: The specimens were collected in wheat field. L98.
Figure 1 [and the subsequent ones]: all the legends must be improved, at present they are a bit obscure.
Response: The legends for all Figures have been improved in this version.
row 185 what is P. musculusall (also row 190)? Why this species and Glyphonys are underlined? Please, carefully check the text for any further misspelling.
Response: Thanks for your corrections. This is indeed a misspelling. We have thoroughly checked the text for any other misspelling.
Figure 6 and 7, the quality is very low, images must be improved.
Response: the quality of Figure 6 and 7 have been improved in this version.